# Effects of Three Months of Detraining on the Health Profile of Older Women after a Multicomponent Exercise Program

**DOI:** 10.3390/ijerph16203881

**Published:** 2019-10-13

**Authors:** Luis Leitão, Ana Pereira, Mauro Mazini, Gabriela Venturini, Yuri Campos, João Vieira, Jefferson Novaes, Jeferson Vianna, Sandro da Silva, Hugo Louro

**Affiliations:** 1Sciences and Technology Department, Superior School of Education of Polytechnic Institute of Setubal, 2910-761 Setúbal, Portugal; anapereiraphd@gmail.com (A.P.); hlouro@esdrm.ipsantarem.pt (H.L.); 2Graduate Program of Physical Education of Cataguases—Sudamerica Faculty, Cataguases 36774-552, Brazil; personalmau@hotmail.com; 3Faculty of Physical Education Sports, Federal University of Juiz de Fora, São Pedro 36036-900, Brazil; reiclauy@gmail.com (Y.C.); joaoguilhermevds@gmail.com (J.V.); jeffnovaes@gmail.com (J.N.);; 4Laboratory of Physical Activity and Health Promotion—Labsau—Postgraduate Program in Sport and Exercise Science from the Institute of Physical Education and Sport, University of Rio de Janeiro State, Rio de Janeiro 21941-599, Brazil; gabriela-venturini@hotmail.com; 5Faculty of Physical Education and Sports, Federal University of Rio de Janeiro, Rio de Janeiro 21941-901, Brazil; 6Studies Research Group in Neuromuscular Responses, University of Lavras, Lavras 37200-000, Brazil; sandrofs@def.ufla.br; 7Sports Science School of Rio Maior, Polytechnic Institute of Santarém, 2040-413 Rio Maior, Portugal

**Keywords:** older adults, detraining, VO_2_, lipidic profile, hemodynamic profile

## Abstract

Physical exercise results in very important benefits including preventing disease and promoting the quality of life of older individuals. Common interruptions and training cessation are associated with the loss of total health profile, and specifically cardiorespiratory fitness. Would detraining (DT) promote different effects in the cardiorespiratory and health profiles of trained and sedentary older women? Forty-seven older women were divided into an experimental group (EG) and a control group (CG) (EG: n = 28, 70.3 ± 2.3 years; CG: n = 19, 70.1 ± 5.6 years). Oxygen uptake (VO_2)_ and health profile assessments were conducted after the exercise program and after three months of detraining. The EG followed a nine-month multicomponent exercise program before a three-month detraining period. The CG maintained their normal activities. Repeated measures ANOVA showed significant increases in total heath and VO_2_ (*p* < 0.01) profile over a nine-month exercise period in the EG and no significant increases in the CG. DT led to greater negative effects on total cholesterol (4.35%, *p* < 0.01), triglycerides (3.89%, *p* < 0.01), glucose (4.96%, *p* < 0.01), resting heart rate (5.15%, *p* < 0.01), systolic blood pressure (4.13%, *p* < 0.01), diastolic blood pressure (3.38%, *p* < 0.01), the six-minute walk test (7.57%, *p* < 0.01), Pulmonary Ventilation (VE) (10.16%, *p* < 0.01), the Respiratory Exchange Ratio (RER) (9.78, *p* < 0.05), and VO_2_/heart rate (HR) (16.08%, *p* < 0.01) in the EG. DT may induce greater declines in total health profile and in VO_2_, mediated, in part, by the effectiveness of multicomponent training particularly developed for older women.

## 1. Introduction

Aging is typically associated with deterioration in mobility and in the performance of activities of daily living, such as dressing, climbing stairs, using the toilet and bathing, which are associated with the loss of muscle mass and decreased maximum rate of oxygen consumption (VO_2 max_) that result from the aging process [1].

To this end, regular physical activity (RPA) is one of the most important actions to maintain a healthy state in the elderly, promoting a slower decline in daily activities performance and improving self-care capabilities [2,3,4]. RPA can also reduce the onset of chronic disease in healthy elderly; improve physical capacities such as resistance, endurance, flexibility and balance performance [3,4]; and help to improve lipidic, glycemic and hemodynamic variables such as blood pressure, triglycerides, cholesterol and glucose [4,5,6]. Further, it can also increase the ability of older people to perform daily tasks, improving quality of life, especially, for older women [6,7,8,9,10].

As the main component of the RPA, exercise training can be applied separately in aerobics programs, resistance programs or through multicomponent programs [11] that can be defined as a program that fully exercises all physical abilities (endurance, muscle strength, coordination, flexibility and balance) in the same way. Recently, multicomponent exercise programs have emerged as an important method in stimulating overall functioning capacity in the elderly [11,12,13,14]. Indeed, a markedly lower capacity for walking or performing daily activities, such as climbing stairs or rising from a chair, may result in impaired cardiorespiratory capacity. 

Furthermore, advancing age in women tends to decrease nutritional status, caused by the deterioration of taste, smell and teeth that can lead to or exacerbate poor nutritional capacity, and consequently increase the likelihood of developing cardiovascular diseases and others glycemic, lipid and hemodynamic parameters [14]. Understanding the cardiorespiratory and metabolic changes in older women after multicomponent training is an important task for this relatively unstudied age group. Determining this influence may promote greater understanding of the potential of varied exercise training focused on total body training to promote therapeutic and functional development in older populations.

Detraining can be considered the partial or total interruption of an exercise program or a partial or total loss of exercise benefits in response to an insufficient training stimulus, with both depending on the duration of training cessation or insufficient training [13,15]. Some studies have described that both metabolic and functional adaptations from exercise programs can decrease even after short detraining periods (DT) due to unexpected causes such as illness and vacation [4,6,13,15,16,17,18]. This fact may result in a reduction in the benefits achieved during the exercise program as functional performance by reducing the physiological capacity. The magnitude of this reduction may depend on training levels previously attained by the subjects or by the length of the period of interruption [12,15]. Lobo et al. [19] conclude that a period of three months of detraining following 1 year of health intervention programs in institutionalized older individuals significantly impairs the majority of the favorable functional changes obtained after training, including agility/dynamic balance, lower body strength and the flexibility of the components of functional fitness—the most affected components of functional fitness.

Little is known in research about the regressive effects of DT in older women and there is not enough data on how DT affects functional fitness changes following the cessation of exercise programs in non-institutionalized elders [20]. Therefore, the purpose of the present study was to analyze the effects of three months of detraining after a nine-month multicomponent training program on cardio-respiratory fitness, lipid, glycemic and hemodynamic profiles in older women. 

We hypothesized that a three-month DT period may induce failures in glycemic, lipid and hemodynamic profiles and significantly decrease cardiorespiratory capacity.

## 2. Materials and Methods 

### 2.1. Sample

Forty-seven functionally independent Caucasian women (aged 60–75 years old) volunteered for the study and underwent a medical evaluation in order to attend the program. The exclusion criteria were: (a) metallic prosthesis implants; (b) osteoarticular dysfunction that could interfere with the execution of the proposed motion; (c) heart problems where the exercise prescription injures the health of the person; and (d) medical contraindication. 

The experimental protocol was approved by the local institutional Ethical Committee for Human Experiments and was performed in accordance with the Declaration of Helsinki. In addition, all subjects signed an informed consent form prior to data collection. Coffee, tea, alcohol and tobacco consumption was prohibited for 24 h before experimental procedures as well as strenuous exercise. 

### 2.2. Procedures

To investigate the effects of a three-month DT period, older Caucasian women were stratified into two homogeneous groups: an experimental group (EG) that carried out a nine-month multicomponent exercise program before the detraining period and a control group (CG) that did not perform exercise (Table 1). 

Data were collected by the same examiner and under the same environmental conditions (10:00–12:00 h; 22–24 °C; 55–65% humidity) at the beginning of the multicomponent program and nine months later (at the end of the multicomponent exercise program; at the beginning of detraining—BD) and after three months of detraining (after detraining—AD). 

#### 2.2.1. Multicomponent Exercise Program

The exercise program consisted of 86 sessions (45 min; twice/week over 9 consecutive months; October–June) of cardio-respiratory fitness, resistance, flexibility and balance as the main components of the program [6]. Each training period consisted of group sessions of aerobic and muscle resistance training, including music appropriate to the activity, age and interests of the older women. The structure of the sessions was: 

(1) A total of 5–8 min of global warm-up activity, including slow walk, calisthenics and stretching exercises. 

(2) A total of 15–25 min of cardio-respiratory workout (aerobics choreography with moderate intensity); the intensity was maintained at 2–3 according to the adapted Borg Rating of Perceived Exertion scale (RPE) in the first month. Subsequently, the intensity was gradually increased up to 4–5 in the adapted Borg RPE.

(3) A total of 15–20 min of resistance training (exercises for upper and lower body). Exercises were performed in a circuit, involving exercises for the upper and lower body, exercises of agility, mobility, coordination and social interaction, with a 30 to 60 s rest period between sets. Weight resistance training was performed using body weight (open and closed kinetic chain exercises) and elastic bands. In order to allow proper familiarization with the exercises with the correct and safe technique of execution and breathing, the training intensity was progressive, especially in the first month. The repetitions and series were increased month after month from 16 to 30 repetitions and from 2 to 4 series.

(4) A total of 5–10 min of relaxation techniques and stretching for the upper and lower body. The flexibility training included static and dynamic stretching techniques.

#### 2.2.2. Detraining

Following the multicomponent training, EG and CG subjects were instructed to carry on their normal lifestyles including dietary patterns and physical routines and avoid any type of systematic exercise for three months (July to September). During detraining, subjects were contacted systematically by phone and in person to ensure that they were not engaged in regular exercise.

#### 2.2.3. Glycemic and Lipid Profile

To measure total cholesterol (TC, mg/dL), triglycerides (TG, mg/dL) and blood glucose (GL, mg/dL) were used a Cobas Accutrend Plus (Roche Diagnostics GmbH, Mannheim, Germany) in accordance with the procedures of Diabetes Atlas Committee (DAC). To carry out blood collection, we used the pen puncture Accu-Chek Softclix ^®^ Pro and their lancets, graded from 1 to 3 in increasing degree of penetration depth, on the distal phalanx palmar of the third finger of the right hand. 

#### 2.2.4. Hemodynamic Profile

The resting heart rate (HR_rest_, bpm) and the blood pressure—systolic pressure (SP, mmHg) and diastolic pressure (DP, mmHg)—measurements were performed with a digital sphygmomanometer for the elders, Omron Digital Blood Pressure Monitor HEM-907 (Omron Healthcare Europe BV, Matsusaka, Japan). These measurements were peformed three times in a sitting position and with the left arm in the support, with 5 min intervals [21]. We also used a scale, OMRON BF 303 (OMRON Healthcare Europe BV, Matsusaka, Japan), with a stadiometer (Seca, Hamburg, Germany) and bioelectrical impedance analysis to assess the weight (kg), height (cm), body mass index (BMI, kg.m^−2^) and body fat percentage (%BF, %).

#### 2.2.5. Cardiorespiratory Fitness

To analyze the cardiorespiratory fitness, the six-minute walk test (6MWT, m) was used with Cosmed K4b2 (Cosmed, Rome, Italy) to measure VO_2_; absolute (VO_2_, mL/min) and relative (VO_2_, mL/kg/min) oxygen uptake; Pulmonary Ventilation (VE, L/min); Metabolic equivalents (METS) (Physiological measure expressing the energy cost of physical activities, defined as the ratio of metabolic rate during the 6MWT to a reference metabolic rate); the Respiratory Exchange Ratio (RER); and the Oxygen Pulse (VO_2_/HR, mL/bpm). Furthermore, a polar cardiofrequencimeter system was used to measure the heart rate (HR, bpm) during the 6MWT.

### 2.3. Statistical Analysis

Data analysis was performed with SPSS 19.0 for Windows (SPSS Inc., Chicago, IL). Descriptive procedures of central tendency and dispersion were used to characterize the variable values and the normality of our sample was verified by the Shapiro–Wilk test. Mixed-model ANOVA was used to examine differences within and between groups over the detraining period.

The delta percentage (∆%) was calculated via the standard formula: ∆% = [(posttest score − pretest score)/pretest score] × 100. The meaningfulness of the outcomes was estimated through the effect size (ES, Cohen’s d, means divided by the standard deviation): 0.2 or less is a small ES, approximately 0.5 is a moderate ES and 0.8 or more is a large ES [22]. For all statistical procedures, the statistical significance accepted was *p* ≤ 0.05.

## 3. Results

All the EG participants completed the nine-month multicomponent exercise program with an attendance rate of 92%. There were no differences among the groups at baseline for age, height, weight or BMI, and only the EG improved glycemic, lipidic and hemodynamic profiles after the exercise program (Table 2). At the start of the detraining period, the EG differed significantly from the CG for hemodynamics and lipid profiles and cardiorespiratory fitness (Table 3).

## 4. Discussion

The aim of our study was to analyze the effects of three months of detraining on cardio-respiratory fitness and lipid, glycemic and hemodynamic profiles of older women, after nine months of a multicomponent exercise program. Our primary finding was that after three months of detraining, older women were unable to maintain the cardio-respiratory fitness, lipid, glycemic and hemodynamic profiles after nine months of regular multicomponent training. However, the older women in the EG comparatively to the CG still have better results than before, showing the positive effect of exercise in the health of the elders.

The decrease in the lipidic, glycemic and hemodynamic profiles found after three months of detraining in the present study agrees with some studies [23,24]. Tokmakidis and Volaklis [24] include in their study on resistance and aerobic training for 8 months followed by three months of detraining, a 3.7% increase in TC and 16.1% increase in TG—results that are similar to ours in TC (4.35%) but higher in TG (3.89%). This difference can be caused by the higher values of TG at the beginning of detraining. These negative effects may be related to the weight and BMI gains observed in the EG, since some studies [19,25] report that some beneficial changes in lipid profiles with exercise training were attributed or correlated with weight loss and to reversed training-induced muscular and biochemical adaptations. Therefore, with our results, we recommend that older women should engage in regular exercise to prevent metabolic syndrome and improve their health status.

Nolan et al. [23]’s study included 4 weeks of detraining after 13 weeks of training and they report changes in %BF (11.62%) and TG (21.74%), but not in SP, DP and TC. Elliot et al. [25] report a 9.32% increase in SP and 1.52% in DP—results that are greater than ours, with a 4.13% increase in SP and 3.38% in DP—after eight weeks of resistance training followed by eight weeks of detraining. These higher results are a result of the lower blood pressure observed after the resistance training program. These increases in SP might be related to the loss of training-induced muscular and biochemical adaptations and to the aging process that developed progressive stiffening of their arterial tree as with age, which leads to a continuous elevation in SP that increases left ventricular work and the risk of left ventricular hypertrophy and, at specific levels, increases the risk of developing coronary heart disease. The increase in DP may only be related to the loss of training-induced muscular and biochemical adaptations, weight gains and a deteriorated lipidic profile, since DP remains normal or decreases with aging [26]. So, our results revealed that three months of detraining can increase the risk for cardiovascular diseases, showing that a multicomponent exercise program should be undertaken permanently without interruptions for the better health of older women.

Few studies have examined the effect of detraining on the cardiorespiratory fitness of older individuals after a multicomponent exercise program [17,20,27]. From a functional perspective, the VO_2max_ has been recognized as an indicator of aerobic capacity [28] and it is well described as one of the most affected by ageing, with progressive deterioration between 0.6% and 2.5% per year. Furthermore, cardiorespiratory capacity is widely associated with performing different activities of daily living [29]. In our study, the significant results of the EG before the detraining period compared to the CG may result from the nine months of multicomponent exercise program [17,30,31]. Nevertheless, the effects of exercise in older people depend on the duration of the program, sample size, initial levels of physical activity and the training protocol [16,29]. Moreover, the duration of the detraining period could be a possible explanation for the discrepancy between the results in older people [16].

In the present study, after the detraining period, all the cardio-respiratory fitness parameters decreased in both groups, although the results for the EG remained higher and showed a smaller decrease than the sedentary adults of the CG. These negative effects are similar to the values reported by Yázigi et al. [32], with a decline in VE (12.8%, 53.7 L/min to 46.8 L/min), VO_2_ (9.9%, 1325.1 mL/min to 1193.3 mL/min), VO_2_/Kg (9.5%, 19.9 mL/min.kg to 18 mL/min.kg), HR (6.1%, 131 bpm to 123 bpm), RER (10%, 1 to 0.9), VO_2_/HR (5.9%, 10.1 mL/bpm to 9.5 mL/bpm) and 6MWT (6.7%, 660.5 m to 616.1 m) after three months of detraining in older individuals (±72 years) following nine months of a multicomponent exercise program. The loss of training-induced muscular and biochemical adaptations of the multicomponent program can be one of the reasons for this decrease in the EG. However the CG did not undergo any exercise programs and but also showed a decrease in their cardio-fitness parameters, and one of the reasons for this might be the season of the year when this detraining period occurred, which is summer and is when the temperature is much higher than other season of the year and this can interfere in the daily routine of the subjects by increasing the inactivity of the subjects.

With the negative impact of detraining on cardiorespiratory fitness, the functional capacity decreased and may also affect the cognitive function of the older people [33]. Barnes et al. [33] reported that cardiorespiratory fitness is positively associated with the preservation of cognitive function over a 6 year period with exercise and may protect against cognitive dysfunction in older people. This cardiorespiratory fitness loss will affect autonomy and quality of life, showing that sedentary behaviors should be replaced alongside active lifestyles, where multicomponent exercise will interact in a positive way with the health and quality of life of older women.

In accordance with recent studies and the present data [11,13,15,17,31], these changes may have significant consequences for older individuals on their health and quality of life by increasing the risk of cardiovascular disease and reducing functional capacity, suggesting that exercise training focused on a multicomponent program has the potential effect of protecting participants against health declines associated with age and that detraining periods should be avoided.

As a limitation of this study, we could highlight that there was no assessment of the subjects’ dietary regimens during the investigation as well as the imbalance in the sample group sizes.

## 5. Conclusions

Three months of detraining is sufficient to cause a decline in the lipidic, glycemic and hemodynamic profile of active older women but even after this detraining period, these women presented better health and quality of life compared to sedentary individuals. We recommend that detraining periods should be avoided and that exercise should be engaged permanently by older women to achieve a better quality of life and health.

## Figures and Tables

**Table 1 ijerph-16-03881-t001:** Subject’s anthropometric characteristics.

Variable	Group EG (n = 28; 70.3 ± 2.3 years) CG (n = 19; 70.1 ± 5.6 years)	Before Exercise (BE)	Beginning of DT	After Three Months of DT	BD vs. AD			
					Confidence Interval		ES	*p*
					Lower	Upper		
Body weight (kg)	EG	71.72 ± 9.24	65.59 ± 8.14 *	66.76 ± 8.18 ^+^	−1.41	−0.42	0.02	0.001
	CG	70.53 ± 7.7	70.31 ± 7.7	71.22 ± 7.72 ^+^	−0.99	−0.30	0.12	0.001
BMI (kg.m^−2^)	EG	29.12 ± 3.91	26.61 ± 3.60 *	27.88 ± 3.59 ^+^	−0.41	−0.12	0.08	0.001
	CG	29.72 ± 3.43	29.65 ± 3.74	30.03 ± 3.68 ^+^	−0.58	−0.17	0.10	0.001
Height (m)	EG	1.57 ± 0.06	1.57 ± 0.06	1.57 ± 0.06				
	CG	1.54 ± 0.04	1.54 ± 0.04	1.54 ± 0.04				

Data presented are the mean ± SD, confidence interval of the difference between BD and AD, effect size (ES) and the p-value of the detraining effect before the nine-month multicomponent exercise program (BE), at the beginning of detraining (BD) and after three months of detraining (AD) of the body mass index (BMI), body weight (kg) and height (cm); * *p* < 0.05, BE vs. BD; ^+^
*p* < 0.05, BE vs. AD. Experimental group (EG); control group (CG).

**Table 2 ijerph-16-03881-t002:** Percentage change in lipidic, glycemic and hemodynamic profiles and cardiorespiratory fitness after the nine-month multicomponent exercise program.

	Group	%BF (%)	SP (%)	DP (%)	HRrest (%)	TG (%)	TC (%)	GL (%)	6MWT (%)
**Before Exercise (BE)** **vs.** **Beginning** **Detraining (DT)**	EG	−15.17 *	−3.80 *	−5.48 *	−9.04 *	−7.64 *	−5.46 *	−10.58 *	11.06 *
	CG	−3.60	−1.71	−1.05	−1.46	−2.53	1.74	−4.03	4.81 ^+^

Data presented are the delta percentages (∆%) before the nine-month multicomponent exercise program (BE) and at the beginning of detraining (BD) of body fat percentage (%BF), systolic pressure (SP), diastolic pressure (DP), resting heart rate (HRrest), triglycerides (TG), total cholesterol (TC), blood glucose (GL), six-minute walk test (6MWT); * *p* < 0.01, BE vs. BD; ^+^
*p* < 0.05, BE vs. BD.

**Table 3 ijerph-16-03881-t003:** Comparison between the variables of the lipidic, glycemic and hemodynamic profiles and cardiorespiratory fitness at the beginning of detraining (BD) and after three months of detraining (AD) in the exercise group (EG) and the control group (CG).

	CG						EG								
	BD	AD	BD vs. AD				BD	AD	BD vs. AD		
			Confidence Interval		ES	*p*			Confidence Interval		ES	*p*
			Lower	Upper					Lower	Upper		
%BF (%)	34.81 ± 3.70	35.69 ± 3.22	−1.87	0.10	0.25	0.043	31.75 ± 3.08 ^¥^	33.60 ± 2.91 ^¥^	−2.30	−1.40	0.62	0.001
SP (mmHg)	140.47 ± 7.24	143.21 ± 6.78	−4.53	−0.94	0.39	0.061	132.11 ± 9.42 ^¥^	136.57 ± 10.14 ^¥^	−6.14	−2.80	0.46	0.001
DP (mmHg)	75.11 ± 8.29	76.95 ± 6.87	−3.72	0.40	0.24	0.065	71.86 ± 4.50 ^¥^	74.82 ± 3.70 ^¥^	−3.73	−2.20	0.72	0.001
HRrest (bpm)	70.16 ± 7.34	71.63 ± 7.53	−2.79	−0.16	0.2	0.081	66.61 ± 6.40 ^¥^	70.04 ± 6.86 ^¥^	−4.57	−2.29	0.52	0.001
TG (mg/dL)	179.89 ± 19.07	184.89 ± 17.26	−13.13	3.13	0.27	0.212	171.46 ± 16.95 ^¥^	178.93 ± 15.88 ^¥^	−10.38	−4.55	0.46	0.001
TC (mg/dL)	186.68 ± 12.97	190.42 ± 16.37	−7.36	−0.12	0.25	0.144	179.00 ± 11.55 ^¥^	185.96 ± 10.50 ^¥^	−10.40	−3.53	0.63	0.001
GL (mg/dL)	84.74 ± 6.79	86.68 ± 7.85	−6.53	−1.36	0.26	0.076	80.71 ± 6.91 ^¥^	82.81 ± 6.53 ^¥^	−6.22	−1.78	0.31	0.001
6MWT (m)	597.89 ± 55.46	567.11 ± 57.81	18.05	43.52	0.54	0.001	658.21 ± 52.55 ^¥^	608.36 ± 60.01 ^¥^	33.80	60.13	0.88	0.001
VE (L/min)	35.59 ± 5.10	30.60 ± 4.62	3.04	6.94	1.03	0.001	44.01 ± 4.42 ^¥^	39.54 ± 4.93 ^¥^	3.18	5.77	0.96	0.001
VO_2_ (mL/min)	1112.49 ± 129.58	1000.34 ± 122.20	78.64	145.66	0.89	0.001	1526.36 ± 192.80 ^¥^	1399.21 ± 193.43 ^¥^	76.11	178.19	0.66	0.001
VO_2_/KG [mL/(min.Kg]	17.50 ± 1.68	15.62 ± 1.71	1.52	2.24	1.11	0.001	21.58 ± 2.35 ^¥^	19.57 ± 2.32 ^¥^	1.24	2.78	0.86	0.001
HR (bpm)	121.50 ± 10.89	116.56 ± 8.00	0.17	9.71	0.52	0.064	132.51 ± 14.49 ^¥^	122.79 ± 13.89 ^¥^	5.66	13.79	0.68	0.001
RER	0.88 ± 0.07	0.81 ± 0.23	−0.044	0.19	0.41	0.212	0.92 ± 0.13 ^¥^	0.83 ± 0.08	0.04	0.12	0.83	0.001
VO_2_/HR (mL/bpm)	10.03 ± 2.27	8.66 ± 2.08	0.82	1.91	0.63	0.001	13.11 ± 4.93 ^¥^	11.00 ± 1.90 ^¥^	0.20	4.01	0.57	0.031
METS	5.11 ± 0.61	4.65 ± 0.64	0.52	0.84	0.74	0.001	5.97 ± 1.17 ^¥^	5.59 ± 0.67 ^¥^	−0.01	0.77	0.4	0.041

Data presented are the mean ± SD, confidence interval of the difference between BD and AD, effect size (ES), p-value of the detraining effect; at the beginning of detraining (BD) and after three months of detraining (AD) of body fat percentage (%BF), systolic pressure (SP), diastolic pressure (DP), resting heart rate (HRrest), triglycerides (TG), total cholesterol (TC), blood glucose (GL), six-minute walk test (6MWT), Pulmonary Ventilation (VE), VO_2_, VO_2_\KG, Respiratory Exchange Ratio (RER), Heart Rate (HR), Oxygen Pulse (VO_2_/HR) and Metabolic equivalents (METS). ^¥^
*p* < 0.05, better results in EG than CG after and before the detraining period.

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
