# Peer review of "Effects of Three Months of Detraining on the Health Profile of Older Women after a Multicomponent Exercise Program"

_ijerph, 2019, doi:10.3390/ijerph16203881_

Round 1

Reviewer 1 Report

The paper is very well structured, and exalts the importance of physical activity in this population, and so it proves that these individuals can´t spend so much time without physical activity. The factor that is a little fragile is how the authors ensure that elderly did not engage in any other PA, how they contacted them??

Author Response

Dear Reviewer,

We are grateful for your consideration of this manuscript, and we also very much appreciate your suggestions, which have been very helpful in improving the manuscript. We also thank the reviewers for their careful reading of our text. All the comments we received on this study of all reviewers have been attended into account in improving the quality of the article, and we present our reply to each of them separately.

POINT 1: The paper is very well structured, and exalts the importance of physical activity in this population, and so it proves that these individuals can´t spend so much time without physical activity. The factor that is a little fragile is how the authors ensure that elderly did not engage in any other PA, how they contacted them??

DONE: The older women were contacted by phone and in person. They were instructed to do their normal daily routines without doing any kind of exercise. (LINE 144)

Reviewer 2 Report

Introduction: this section is well structured. The author(s) presents a theoretical framework supported by relevant and current literature to explain the importance of the study. The objective and hypotheses are well specified and located in the document.

Method: The way of recruiting the participants as well as the methodology of the study seem adequate to respond to the hypotheses raised.

Results: They are presented in an orderly manner and are interesting for the field of study.

Discussion: In this section the author(s) explain the data obtained and their relationship between the variables analyzed.

In conclusion, the study seems adequate and the obtained results bring valuable information to the field of study. However, I would like to make some suggestions for improvement:

have the document reviewed by a native speaker. explain the mechanism of detraining in the variables analyzes include a practical application section, proposing interventions that can serve to avoid non-exercise at those ages, as https://www.sciencedirect.com/science/article/abs/pii/S0031938418310370  

Author Response

Dear Reviewer,

We are grateful for your consideration of this manuscript, and we also very much appreciate your suggestions, which have been very helpful in improving the manuscript. We also thank the reviewers for their careful reading of our text. All the comments we received on this study of all reviewers have been attended into account in improving the quality of the article, and we present our reply to each of them separately.

POINT 1: have the document reviewed by a native speaker. explain the mechanism of detraining in the variables analyzes include a practical application section, proposing interventions that can serve to avoid non-exercise at those ages, as https://www.sciencedirect.com/science/article/abs/pii/S0031938418310370  

DONE: We add a definition of detraining and his side effects (LINE 71) and include some recommendations to avoid non-exercise and sedentarism of the olders (LINE 217; 230 and 261)

Reviewer 3 Report

Effects of 3-month detraining on health profile of older women after a multicomponent exercise program

General

Overall, this is a well written study with a novel research question of detraining. I would like to commend the authors for the effort to improve the knowledge in detraining and especially in older women. Authors establish a novel research question and found that 3-months detraining period significant declines total health profile and in VO2max in older women. In spite the fact that the findings are encouraging and pleasant, there are some points that need to be addressed before the study proceeds for publication.

Basic corrections and comments

Authors devoted a lot of text to the 9 months training program and to the increases that this multi-component training program induced to the older participants. However, inside the manuscript these changes of pre to post measurements from the 9 months multicomponent training program are absent. Having this information would be helpful for the readers to observe and compare the initial values of the health profile and VO2max before the 9 months training.

Abstract need to be re-written. Although authors present the aim, the methods and the main finding, the existence of many abbreviations inside the results make the abstract difficult to understand.

As a final point, authors should provide test re-test reliability for all measurements and performance tests.  

Specific

Abstract

Abstract provides many abbreviations which have not been presented. Thus, I suggest to authors to re-write the Abstract in order to become better in reading and understanding.

Line 30: Change 47 number to Forty seven.

Introduction

The introduction is well written and provides good and new information on detraining and exercise in older people. However, I lacked some information about the effects of multi-component exercise or just exercise on lipid, glycemic and hemodynamic profiles of older individuals. I suggest to authors to add some references to support the effect of exercise in these components.

Line 56: Please clarify “muscle resistance”. Is this muscle strength (general strength, muscle hypertrophy or/and muscle endurance?)

Line 64-66: Authors have already concluded the 9 months multicomponent training program. Someone would expect to find the results of this training program inside this paper.

Line 69: I suggest to authors to provide the definition of detraining before entering the rest of the paragraph. What is detraining for older women?

Methods

Table 1: It would very interesting to present the anthropometric measurement in the beginning of the 9 months training program. Was there any difference between this time measurement and after the end of detraining?

Paragraph 2.2.1: Here I am confused. Authors present the 9 months multi-component training program but they don’t present the results from this. Pre to post values comparisons and comparisons between pre and detraining values. I agree with the existence of the control group, but it would be interesting to examine these changes and whether the training program affected the health profile and VO2max in the same experimental group. In short, there is no reason to present the 9 months multi-component training program if authors don’t provide the results.

Paragraph 2.2.3, 2.2.4 and 2.2.5: Please provide intraclass-correlation coefficient (ICC, upper – lower bound) for all measurements.  

Line 175: I suggest to authors to add a reference.

Results

Line 181: Was there a difference before the 9 months multicomponent program? How much the experimental group improved?

Discussion

The section of discussion is very easy to read and provides good comparison with other studies. However, authors have to provide a final take home message from their study. Is detraining something that older women can follow? Can the authors suggest a period of detraining where no loses in health profile and VO2max will occur? Was that multi-component training program good for all the participants? Can it be applied in older men?

Conclusions

Line 252: Please clarify “after this training program” …

Author Response

Dear Reviewer

We are grateful for your consideration of this manuscript, and we also very much appreciate your suggestions, which have been very helpful in improving the manuscript. We also thank the reviewers for their careful reading of our text. All the comments we received on this study of all reviewers have been attended into account in improving the quality of the article, and we present our reply to each of them separately.

Point 1: Basic corrections and comments

Authors devoted a lot of text to the 9 months training program and to the increases that this multicomponent training program induced to the older participants. However, inside the manuscript these changes of pre to post measurements from the 9 months multicomponent training program are absent. Having this information would be helpful for the readers to observe and compare the initial values of the health profile and VO2max before the 9 months training.

DONE: We add table 2 with delta percentage (∆%) of the lipidic, glycemic and hemodynamic profile changes of pre vs post measurements from the 9 months multicomponent training program (LINE 189).

Abstract need to be re-written. Although authors present the aim, the methods and the main finding, the existence of many abbreviations inside the results make the abstract difficult to understand.

DONE: We add all the non-common abbreviations in full extension (LINE 37) 

As a final point, authors should provide test re-test reliability for all measurements and performance tests.

DONE:  It´s not possible to perform a test re-test because all data were already collected. However all procedures done in the assessments are validated, and done in the same conditions and with the same examiner (Line 113)

POINT 2: Specific

Abstract

Abstract provides many abbreviations which have not been presented. Thus, I suggest to authors to re-write the Abstract in order to become better in reading and understanding.

Line 30: Change 47 number to Forty seven.

 DONE: We add all the non-common abbreviations in full extension (LINE 37) and changed 47 to forty-seven (LINE 31).

Introduction

The introduction is well written and provides good and new information on detraining and exercise in older people. However, I lacked some information about the effects of multi-component exercise or just exercise on lipid, glycemic and hemodynamic profiles of older individuals. I suggest to authors to add some references to support the effect of exercise in these components.

DONE: We add some benefits of multicomponente exercise and exercise with some references. LINE 53 - 54

Line 56: Please clarify “muscle resistance”. Is this muscle strength (general strength, muscle hypertrophy or/and muscle endurance?)

DONE: We accepted the suggestion and changed to muscle strength. (Line 58)

Line 64-66: Authors have already concluded the 9 months multicomponent training program. Someone would expect to find the results of this training program inside this paper.

Line 69: I suggest to authors to provide the definition of detraining before entering the rest of the paragraph. What is detraining for older women?

DONE: We add a definition of detraining and his side effects (LINE 71) 

Methods

Table 1: It would very interesting to present the anthropometric measurement in the beginning of the 9 months training program. Was there any difference between this time measurement and after the end of detraining?

DONE: We add the data in table 1 (LINE 108).

Paragraph 2.2.1: Here I am confused. Authors present the 9 months multi-component training program but they don’t present the results from this. Pre to post values comparisons and comparisons between pre and detraining values. I agree with the existence of the control group, but it would be interesting to examine these changes and whether the training program affected the health profile and VO2max in the same experimental group. In short, there is no reason to present the 9 months multi-component training program if authors don’t provide the results.

DONE - We add table 2 with delta percentage (∆%) of the lipidic, glycemic and hemodynamic profile changes of pre vs post measurements from the 9 months multicomponent training program (LINE 189).

Paragraph 2.2.3, 2.2.4 and 2.2.5: Please provide intraclass-correlation coefficient (ICC, upper – lower bound) for all measurements.  

DONE: It’s not possible to provide because we did not made a test re-test for all measurements (only one test in each point assessment)

Line 175: I suggest to authors to add a reference.

DONE: We add a reference (Cohen, 1988) (LINE 181)

 Results

Line 181: Was there a difference before the 9 months multicomponent program? How much the experimental group improved?

DONE: We add table 2 with delta percentage (∆%) of the lipidic, glycemic and hemodynamic profile changes of pre vs post measurements from the 9 months multicomponent training program (LINE 189).

Discussion

The section of discussion is very easy to read and provides good comparison with other studies. However, authors have to provide a final take home message from their study. Is detraining something that older women can follow? Can the authors suggest a period of detraining where no loses in health profile and VO2max will occur? Was that multi-component training program good for all the participants? Can it be applied in older men?

DONE- We add some recommendations during discussion from our study results of older women to avoid non-exercise and sedentarism of the olders (LINE 217; 230 and 261)

 Conclusions

Line 252: Please clarify “after this training program” …

DONE: We re-write the conclusion (LINE 272)

Round 2

Reviewer 3 Report

No comment